# Homologous Recombination Deficiency (HRD) Scoring, by Means of Two Different Shallow Whole-Genome Sequencing Pipelines (sWGS), in Ovarian Cancer Patients: A Comparison with Myriad MyChoice Assay

**DOI:** 10.3390/ijms242317095

**Published:** 2023-12-04

**Authors:** Giovanni L. Scaglione, Sandro Pignata, Angela Pettinato, Carmela Paolillo, Daniela Califano, Giuseppa Scandurra, Valentina Lombardo, Francesca Di Gaudio, Basilio Pecorino, Liliana Mereu, Paolo Scollo, Ettore D. Capoluongo

**Affiliations:** 1Laboratory of Molecular Oncology, IDI-IRCCS, Via dei Monti di Creta, 104, 00167 Rome, Italy; gianluca.scaglione@gmail.com; 2Department of Urology and Gynecology, Istituto Nazionale Tumori IRCCS Fondazione G. Pascale, Via Mariano Semmola, 53, 80131 Naples, Italy; s.pignata@istitutotumori.na.it; 3Department of Pathological Anatomy, A.O.E. Cannizzaro, Via Messina, 829, 95126 Catania, Italy; apettinato20@gmail.com; 4Department of Clinical and Experimental Medicine, University of Foggia, Viale Luigi Pinto, 71122 Foggia, Italy; carmela.paolillo@gmail.com; 5Functional Genomic Unit, Istituto Nazionale Tumori IRCCS Fondazione G. Pascale, Via Mariano Semmola, 53, 80131 Naples, Italy; d.califano@istitutotumori.na.it; 6Department of Medical Oncology, A.O.E. Cannizzaro, Via Messina, 829, 95126 Catania, Italy; giusy.scandurra@gmail.com (G.S.); lombardovalentina89@gmail.com (V.L.); 7A.O.U.P. Paolo Giaccone, Via del Vespro, 129, 90127 Palermo, Italy; 8Department of Obstetrics and Gynecology, A.O.E. Cannizzaro, Via Messina, 829, 95126 Catania, Italy; eliopek@gmail.com (B.P.); paolo.scollo@unikore.it (P.S.); 9Division of Obstetrics and Gynecology, Department of General Surgery and Medical-Surgical Specialism, University of Catania, P.O. “G Rodolico”, Via Santa Sofia, 78, 95123 Catania, Italy; mereu.lilly@gmail.com; 10Faculty of Medicine, “Kore” University, Cittadella Universitaria, 94100 Enna, Italy; 11Department of Clinical Pathology and Genomics, A.O.E. Cannizzaro, Via Messina 829, 95126 Catania, Italy; 12Department of Molecular Medicine and Medical Biotechnology, Federico II University, Via Pansini, 5, 80131 Naples, Italy

**Keywords:** academic HRD, sWGS, PFS, HGSOC

## Abstract

High-grade serous ovarian cancer (HGSOC) patients carrying the *BRCA1/2* mutation or deficient in the homologous recombination repair system (HRD) generally benefit from treatment with PARP inhibitors. Some international recommendations suggest that *BRCA1/2* genetic testing should be offered for all newly diagnosed epithelial ovarian cancer, along with HRD assessment. Academic tests (ATs) are continuously under development, in order to break down the barriers patients encounter in accessing HRD testing. Two different methods for shallow whole-genome sequencing (sWGS) were compared to the reference assay, Myriad. All these three assays were performed on 20 retrospective HGSOC samples. Moreover, HRD results were correlated with the progression-free survival rate (PFS). Both sWGS chemistries showed good correlation with each other and a complete agreement, even when compared to the Myriad score. Our academic HRD assay categorized patients as HRD-Deficient, HRM-Mild and HRN-Negative. These three groups were matched with PFS, providing interesting findings in terms of HRD scoring and months of survival. Both our sWGS assays and the Myriad test correlated with the patient’s response to treatments. Finally, our AT confirms its capability of determining HRD status, with the advantage of being faster, cheaper, and easier to carry out. Our results showed a prognostic value for the HRD score.

## 1. Introduction

Ovarian cancer (OC) is the most lethal gynecological cancer, and the eighth leading cause of cancer-related death among women worldwide [1,2]; the WHO estimates that every year 225,500 new cases will be diagnosed [2]. During the last decade, the advance in treatment for ovarian cancer led to a high improvement in overall survival (OS), mainly due to the use of angiogenesis inhibitors and poly-ADP ribose polymerase (PARP) inhibitors [1], but despite this improvement, the OS remains low, at 40% for stage III and 20% for stage IV [3]. Nowadays, it is known that OC is not a unitary disease, but it needs to be classified as one of multiple distinct entities sharing the same anatomical site upon presentation [2]. The WHO classified this neoplasm into five subtypes, based on immunohistochemistry, namely, high-grade serous (HGSOC), low-grade serous (LGSC), endometrioid (EC), clear cell (CCC) and mucinous cancers (MC), each of which shows different behavior and response to treatment [3]. It was estimated that BRCA mutation in the general population is about 1% and germline mutation in these genes take part in the development of 10–20% of epithelial ovarian cancer (EOC), usually of the high-grade serous subtype [2,3]. Lots of work has been carried out to define the HGSOC molecular classification and to identify the biology and molecular features of each subtype [4,5]. Genetic factors represent one of the major driving causes of OC onset, where the tumor-suppressor BRCA1 and BRCA2 gene alterations are the most associated with the increased cancer susceptibility [3].

The standard treatment for EOC includes surgery and platinum-based chemotherapy, but BRCA1/2 mutation carriers respond well to platinum-based chemotherapy and (PARP) inhibitor treatment [5]. Different types of cancer show a germline BRCA1/2 mutation (mainly ovaries, breasts, prostate, and pancreas), and a consequent high sensitivity to DNA-damaging drugs such as platinum, doxorubicin, and topoisomerase inhibitors [6].

The BRCA1/2 proteins play a crucial role in the repair of double-strand breaks of DNA, by a homologous recombination system. In the HGSOC showing a deficiency in this system, drugs targeting PARP1/2 enzymes have a strong anti-tumor activity, due to a mechanism known as synthetic lethality [2]. Thanks to this evidence, in 2005, the US Food and Drug Administration approved PARP inhibitors for BRCA-associated cancers [6].

It is known that treatment with PARP inhibitors of HGSOC patients carrying a BRCA1/2 mutation led to a prolonged progression-free survival (PFS): nevertheless, some patients without these alterations showed similar benefits from the same treatment [4].

Consequently, the identification of patients deficient in the homologous recombination repair (HRR) system (namely, HRD), who could benefit from this treatment, still represents a clinical unmet need [7]. The alteration in one of the other HRR factors led to cells that show features like that of the BRCA1/2 mutation carrier: this status, defined as “BRCAness”, is also related to a greater risk of developing breast and ovarian cancer [8].

The definition of the overall molecular signatures surrounding the HRD status (through the evaluation of mutational status in other components of DNA repair pathways, or methylation of the BRCA1 promoter or specific genomic scar scoring), could lead to the better selection of patients who benefit from PARP inhibition [4].

Due to the evidence that, in OC and breast cancers, the HRD score defined by telomere allelic imbalances, large-scale transitions, loss of heterozygosity (LOH), and mutational signature 3, showed higher sensitivity to platinum and PARP inhibitors, even in the absence of BRCA1/2 mutations, the implementation of HRD assessment is becoming mandatory for the clinical management of OC patients [6].

Up until today, several works evaluating the different methodologies available for defining HRD status have been published. Nonetheless, there is not yet a unique assay able to determine HRD status of all cancers potentially eligible for treatment with PARP inhibitors.

After the results of the PRIMA and PAOLA1 trials, the gold standard for the identification of the HRD status is the commercial assay myChoice CDx (Myriad; Myriad Genetics, Salt Lake City, UT, USA) [9].

This test detects the presence of BRCA1 and BRCA2 alterations, along with other parameters (loss of heterozygosity, telomeric-allelic imbalance, and large-scale state transition) [10,11]. However, the access to this assay is limited by many factors, such as high cost and no reimbursement in many countries, along with technical issues from inconclusive and false-negative results. Finally, there are also logistical difficulties related to the need to send samples to a centralized laboratory overseas.

The major issue is the definition of HRD status, which is related to the type of assay performed. For example, the myChoice CDx assay relies on the presence of the *BRCA1/2* mutations, but the mutational status of these genes is not always determined correctly by commercial tests, with evident discrepancies in methodological comparison works [11].

Lately, the MITO (Multicenter Italian Trial in Ovarian Cancer) group has published a work on the development of an academic test able to overcome the difficulties due to the logistics- and access-related issues, which are mainly due to the following: (a) the lack of a harmonized NGS infrastructure in Italy, as in other European countries; (b) the high costs of the outsourced assays; and (c) the differences in the score calculated by the available commercial and outsourced assays, with the need for the coming IVD rules to comply within European countries [12,13]. The results obtained by the Italian MITO group are extremely promising, as shown by the significant high correlation with the Myriad test and patient outcome [11].

The main aim of this work was to compare the academic HRD-MITO assay (aHRD-MITO) as performed using the shallow NGS Roche method (namely, “R”), coupled with the bioinformatics algorithm for HRD deciphering [11], with a new shallow WGS chemistry (provided by Watchmaker; namely “W”), and the myChoice CDx.

The secondary endpoints were the following: (a) the comparison of the analytical performances obtained by R and W sWGS pipelines with the reference assay provided by Myriad [14]; (b) the correlation of the MITO academic genomic assessment with the numerical scores (from 0 to 100) given by Myriad; and (c) the correlation of HRD scores obtained by the R, W and Myriad assays with progression-free survival (PFS), respectively.

## 2. Results

### 2.1. BRCA1/2 and HRD Assessment of Our NGS Pipelines and Myriad

Table 1 shows all genomic findings related to the 20 ovarian cancer patients.

In patient n.1, Myriad did not report the c.4284dupT BRCA2 alteration that was arbitrarily listed within the “non-clinically significant findings” as pathogenic. This variant, which was also confirmed at germline level, not only in the proband but also in family members suffering hereditary cancers, is indeed reported as Class 5 (pathogenic) in all genomic databases (ClinVar and Varsome). In patient n.4, the BRCA1 c.649del, which is reported by the Franklin database (https://franklin.genoox.com/clinical-db/ accessed on 10 September 2023) as Class 4 (likely pathogenic), was indeed annotated by Myriad as a non-clinically significant finding. We underline the fact that the c.649del variant resulted to be exclusively somatic, with a coverage of 40% in our patient. The definitive classification of this variant is still pending.

In patient 7, the VUS c.9052_9057del was not identified by the Myriad assay.

In patient n.10, Myriad did not report as pathogenic the variant BRCA2 c.4284dup, which was indeed carried not only by the proband but also by other family members suffering from hereditary cancers.

In patient n.12, Myriad did not identify the somatic BRCA2 c.8755-1G>A pathogenic variant. This variant was indeed enriched in the same sample by our NGS pipeline, with a VAF of 36% (coverage 4500×).

In patient n.19, Myriad did not identify the BRCA1 c.65T>C mutation that we confirmed also at germline level in this patient. Therefore, the borderline HRD scoring of 41 given by Myriad can be not accurate, due to the missing information regarding BRCA status, while our HRD assessment resulted as to be highly positive.

### 2.2. Academic GIA (aHRD-MITO) and sWGS Library Comparison

Sequencing data obtained from both sWGS kits were used to classify the samples with the aHRD-MITO scores, based on our in-house bioinformatic pipeline. The scores obtained were used to estimate the level of correlation between the two technologies, as well as to stratify the patients in three groups. Based on the large-scale genomic alteration (LGAs), we identified the following: **HRD-Deficient** (LGA > 20), **HRM-Mild** (15 < LGA < 19) and **HRN-Negative** (LGA < 14) [13]. We used the “N” suffix to define the samples which resulted to be negative from using our pipeline. As in the case of the overall HRD NGS-based assays, this means that we cannot be sure that the non-HRD samples are really fully HR-proficient. As we have already published, this proficiency can be assessed only using the functional scores derived from RAD51 foci testing [9,11]. In Table 1, the data regarding the HRD (R + W) classification are also reported. Both the “**R**” and “**W**” **methods** are highly correlated to each other, with a Spearman’s rho coefficient of 0.914 (*p* = 0.000). After patient stratification [Table 2], Cohen’s kappa was calculated, to measure the level of agreement of the two methods when used to generate the HRD score. Substantial agreement between the two methods was established, with a k = 0.736 (*p* = 0.000).

### 2.3. Comparisons between the aHRD-MITO Pipeline and Myriad myChoice^®^ Assay

The aHRD-MITO GIA, calculated using sequencing data collected using both methods, has been compared to the score reported by Myriad myChoice^®^. We obtained a Spearman’s rho coefficient of 0.734 (*p* = 0.000) and 0.699 (*p* = 0.001) for W and R, respectively, when compared to the Myriad score, corresponding to a substantial agreement. In order to compare the results with the Myriad classification, we used the GIA obtained using the two in-house methods, to stratify the patients into two groups: HRD-Deficient (LGA > 15), and HRN-Negative (LGA < 14) (data shown in Table 3).

Seventeen samples had positive HRD results for cutoff ≥42. In contrast, R and W scores were positive in 16 and 17 samples, respectively. The PPA was 94% for R and 88% for W, compared to myChoice. Cohen’s kappa was 0.608 (*p* = 0.007, substantial agreement) and 0.483 (*p* = 0.028, moderate agreement) for R and W, respectively. As expected, the Fisher’s exact test also resulted to be perfectly in agreement with the other statistical tests. The method R showed an overall better level of agreement with the Myriad patients’ classifications. Therefore, method R was used for the further comparisons and data analyses. Finally, we tried to associate our three different scores (HRN, HRM, HRD) with the scores given by Myriad, as shown in Table 4. All Myriad scores below 42 value corresponded to GIA < 14 (resulting as HR-negative), while the highest number of genomic imbalances (>20) were associated with the highest Myriad scores.

### 2.4. Correlation between the aHRD-MITO Score and PFS

Data collected on 20 ovarian cancer patients were used to estimate the progression-free survival rate for HRD, HRM and HRN patients. All GIA scores were calculated from sequencing data obtained using the R method.

The results of the progression-free survival (PFS) univariate analysis utilizing three different LGA thresholds (>20, 15 < LGA < 19, and <14) are shown in Figure 1.

It is noteworthy that our scores significantly correlated with PFS: in fact, patients with the highest HRD scoring showed the longest mean PFS (95.04 months), as compared to the mild (39.97 months) and negative (20.73 months) ones.

## 3. Discussion

In this study, we carried out two methods of shallow whole-genome sequencing (R and W) to identify genomic instability and assess the HRD status in 20 ovarian cancer samples, nineteen classified as HGSOC and one as endometrioid, all collected before administering drug treatment.

When evaluating the R and W aHRD-MITO assays, both shallow NGS methods showed a high correlation with each other, and a complete agreement. Even when compared with the score provided by Myriad, a substantial agreement was found.

Nevertheless, when we stratified the patients into HRD-Deficient and HRN-Negative groups for the comparison with the Myriad classification, the R methods showed an overall better level of agreement, and were therefore used to set up the subsequent analysis.

Based on our results, we were able to assign as negative (HRD-N) those samples below the Myriad score < 42, as HRD-mild those corresponding to Myriad score values between 43 and 55, and as HRD-Deficient those with a Myriad score above fifty-six. These results are representative of the significant level of agreement found in our samples obtained using the same bioinformatics pipeline in the two different sWGS assays. It is noteworthy that, although the aHRD-MITO scoring was assessed without evaluating the *BRCA1/2* status or the molecular signatures (SBS3 and SBS8) [15], we were able to identify different levels of large genomic-scale alterations related to different Myriad score ranges, confirming that our in-house (academic) algorithm was able to define the HRD status.

Furthermore, when the correlation of the HRD classification of the “R” method with progression-free survival (PFS) was performed, patients classified as HRD showed a significantly higher probability of survival, with approximately 95 months of mean PFS, as compared to those classified as HRM (PFS = 39.97) or HRN (20.73). These findings are in agreement with results obtained by other authors in similar contexts [16].

It is important to underline the fact that, although the BRCA1/2 mutational status can contribute to an improvement in different bioinformatics pipelines, it has been recently established, as not all *BRCA1/2* mutations can contribute equally to the HRD feature [17], and *BRCA1/2* somatic testing can also fail to detect pathogenic variants with highly consolidated NGS pipelines [11]. Therefore, it would be likely that patients with a BRCA1/2 mutation not having a severe deleterious effect on HRR-pathway biology could show an indeterminate or negative HRD score, or that patients without any *BRCA1/2* mutation could show an HRD status which is dependent on other mechanisms of HRR impairment (RAD51 loss or methylation of *BRCA1* promoter).

Nevertheless, as recently reported, the co-testing of HRD and *BRCA1/2* germline/somatic testing should be implemented, to enable optimal and timely treatment decisions on maintenance therapy, as well as for testing patients on whom the HRD test will not be evaluable [18].

Furthermore, it has recently been shown that the benefit of combo therapy (olaparib and bevacizumab) is particularly high for patients with mutations located in the BRCA1/2 DNA-binding domain, and is also associated with a good or excellent outcome [17]. Therefore, the independence of our academic HRD pipelines in relation to the BRCA1/2 testing could facilitate the assay implementation: a patient diagnostic pathway based on the HRD screening, followed by BRCA1/2 tumor and germline testing (reflex modality), could significantly reduce the issues related to the errors in BRCA1/2 assessment which are still affecting the different commercial and in-house pipelines [11,19,20,21].

Moreover, our data show that HRD testing can assume both a prognostic and a predictive value, since PFS is strictly correlated with the value of the score.

Finally, we would underline many important issues coming from our study, which could facilitate, in the near future, the access to this type of test, which is still available only from outsourcing or to different alternative methods, whose reimbursement is still challenging for most of the national and regional health systems [12,22].

We consider our results to be really exiting, for the following reasons:(a)Our assessment is completely independent of the BRCA1/2 results. In this regard, although the guidelines for tumor BRCA testing [23] have been published, many studies have recently shown that the rate of failure of somatic BRCA testing still does not fall below 2–5% of the overall samples tested, using the different methodologies routinely used [19,20,21]. The failure is also related to different modalities of variant classification [24,25]. Our paper clearly shows that both the misclassification and the lack of detection of BRCA1/2 mutations in clinical samples can affect the quality of reporting, particularly when the BRCA results are part of the algorithms used for HRD assessment. Moreover, the missing identification of a pathogenic variant in the tissue, particularly when confirmed as germline, can alter the patient and family management, not only in terms of therapy administration, but also for preventive purposes.(b)Our bioinformatics analysis for GIA scoring was independent of the type of sWGS employed, since both “R” and “W” chemistries resulted to be qualified for the large-scale genomic alteration assessment, providing superimposable results, even when compared to the Myriad assay. Thus, our data confirm the utility of using the sWGS pipeline in assessing HRD status.(c)Costs for genomic testing are becoming an important barrier for patients who have some difficulties in covering the costs of this test on their own. Moreover, health systems do not have the capability to sustain financial coverage of overall genomic assays. In this regard, the availability of HRD assays that are cheaper and more accessible to patients is an important goal to achieve, in order to facilitate patient pathway and management. In this regard, our HRD assay is (1) easy to carry out; (2) fast, since it takes only four full days to prepare and run forty-eight samples; (3) cheap, since overall costs for our test are close to EUR 500; and (4) implementable by different hospital laboratories participating in the HRD Italian network within the MITO-group.

Our results are in favor of rapid implementation of the HRD test in the clinical setting, particularly when performed under ISO1589 rules, to respond to the quality assurance and IVDR requirements, particularly for laboratory-developed tests, in the limbo of this transition period of IVDR implementation [12].

To achieve this goal, we should also make some efforts to reduce biases related to the pre-analytical level, where tumor purity has been recently shown to be the most critical factor for the determination of the HRD score [26].

As a limit of our study, we can underline the small cohort of patients analyzed: although we performed the test on only twenty samples in three different modalities, our patients were all fully clinically characterized, also for the follow-up. We want to point out that the HRD MyChoce is very expensive, and costs for the NGS reagents (particularly the high-throughput flow cells) are high; not all research groups can cover costs for these analytical comparisons, outside of clinical trials supported by big pharma, as is the case in the present study.

Finally, the standardization and harmonization of the entire HRD pipeline is becoming mandatory, in order to guarantee the best testing and patient management and, in particular, to match the ISO15189 and IVDR requirements [12].

## 4. Materials and Methods

### 4.1. Patients

Under a protocol approved on the 29 of June 2023 by the Ethical Committee of Enna “Kore” University (prot. N 12147/2023) 20 high-grade serous and endometrioid ovarian-cancer patients who received surgery at the Department of Obstetrics and Gynecology of Cannizzaro Hospital were retrospectively enrolled, in a consecutive series. Cancer tissue specimens were used to assess the HRD genomic instability score (GIS) evaluated with regulatory-approved Myriad myChoice^®^ CDx, in comparison with our previously published in-house academic genomic tests of genetic instability (namely aHRD-MITO) and the “W” pipeline [11]. Data collected from 20 ovarian cancer patients were used to estimate the progression-free survival rate for HRD, HRM and HRN patients. All patients provided informed consent for the use of their data for research purposes, before enrolment. The study was conducted in accordance with the Declaration of Helsinki.

The mutational status of the 20 enrolled patients was as follows: N = 18 BRCA-mutated (11 in BRCA1; 7 in BRCA2); 1 = carrier of BRCA2 VUS; and 1 was negative for BRCA alterations. The types of alterations are listed in Table 1.

### 4.2. DNA Isolation and Shallow Whole-Genome Sequencing (WGS) DNA Libraries

Total genomic DNA from formalin-fixed paraffin-embedded (FFPE) tissues was extracted with the QIAamp DNA FFPE Tissue Kit (QIAGEN, Hilden, Germany), according to manufacturer’s instructions. Shallow WGS DNA libraries for Illumina sequencing were prepared using two different methods: (1) “R”—the Roche shallow WGS KAPA HyperPlus kit, according to the manufacturer’s instructions (Roche Sequencing Solutions, Pleasanton, CA), following the same protocol already published [11]; and (2) “W”—the Watchmaker DNA Library Prep Kit (Watchmaker Genomics, Boulder Colorado), using 100ng of DNA. We performed fragmentation at 37 °C for 20 min, to obtain fragments of approximately 200 bp. The ligation reaction was incubated at 20 °C for 15 min and after that, the cleanup of the ligation was performed immediately. Library amplification was performed to obtain estimated final yields, based on a library-mode insert size of 300 bp, including adapters. The quality and integrity of the libraries were evaluated on the TapeStation (Agilent Technologies, Santa Clara, CA, USA), using the Agilent D1000 ScreenTape System. The concentration of all the libraries was measured using the Qubit dsDNA High Sensitivity (HS) Assay Kit on the Qubit^®^ Fluorometer 4.0 (Invitrogen Co., Life Sciences, CA, USA), and the value was used to convert ng/ul to nM. Two nanomoles of each “W” library was prepared, before pooling all the samples together. The paired-end sequencing reaction was performed on the Illumina NextSeq550 Dx System (Illumina, San Diego, CA, USA), loading the pool with a concentration of 1.5 pM and 2% Phix 1.5 pM.

### 4.3. Clinical- and Genetic-Data Collection

Clinical data were extracted from electronic medical records, including age, FIGO stage, tumor grade, histology, chemotherapy, Overall Survival (OS), and Progression-Free Survival (PFS) [Table 5].

As indicated by our AIOM-Guidelines [26], all these patients underwent BRCA1/2 testing, to assess the eligibility for PARP-1 inhibitors. Although some different methods for BRCA1/2 testing are present on the market [27], we performed through the BRCA1 and BRCA2 full-gene screening, using the DEVYSER BRCA NGS kit (DEVYSER, Hägersten, Sweden), according to the manufacturer’s instructions, as already published [28].

### 4.4. Genomic Instability Assessment (GIA)

GIA was calculated using whole-genome sequencing (WGS) data at low coverage (0.4–0.8×) for each sample, using six different integrated models, encompassing variable sliding windows spanning 5–1000 Kb. Briefly, all fastq files passing quality control were processed and aligned to the hg19 reference genome, using a customized Sarek pipeline [29,30], built using the Nextflow workflow manager [31]. All Bam files passing quality checks were processed in the designed R module of our *in-house* pipeline. At this stage, six independent models spanning the 5-to-1000 kilobase genome window, implemented using the LGA method [14] and a neural network module [32], provided the predicted HRD status, as previously described [11].

### 4.5. Statistical Analysis

Descriptive statistics were used to summarize the demographic and clinical characteristics of the study population. Baseline characteristics of patients were described using the median and interquartile range for continuous variables and frequencies and percentages for the qualitative variables. The level of correlation between the 2 WGS technologies (R and W) used to calculate the aHRD-MITO scores was established using Spearman’s coefficient. Also, the test of association, Fisher’s exact test, was used. The concordance index was measured using Cohen’s k statistic, with 95% CI. The k statistic was interpreted as <0, indicating no agreement, 0.00–0.20 as slight, 0.21–0.40 as fair, 0.41–0.60 as moderate, 0.61–0.80 as substantial, and 0.81–1.00 as an almost perfect agreement. The percentage positive agreement (PPA) among the scores in the entire cohort was also assessed. Overall survival (OS) was measured from date of diagnosis to date of death or last follow-up. The patients who were alive were censored at the date of the last follow-up. PFS was measured from the date of diagnosis to the date of earliest recurrence, progression, or death. Patients were censored on the date of the last evaluation for recurrence or progressive disease. Kaplan–Meier was used to estimate OS and PFS, and results were modeled using the Cox proportional hazards regression as a function of prognostic factors for estimating the hazard ratio, with 95% confidence intervals. All statistical analyses were performed using SPSS software ver. 26.0 (IBM, Armonk, NY, USA).

## 5. Conclusions

The present paper shows that, when sWGS is coupled with a powerful and a clinically validated bioinformatics pipeline, HRD can be easily assessed, regardless of *BRCA1/2* testing. Our data can be helpful in the implementation of the HRD assay in routine clinical settings.

## Figures and Tables

**Figure 1 ijms-24-17095-f001:**
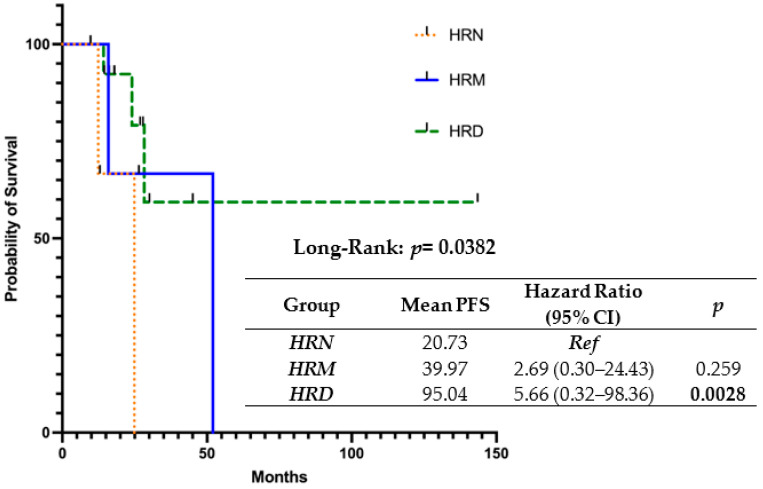
The association between HRD, HRM, HRN groups and progression-free survival.

**Table 1 ijms-24-17095-t001:** *BRCA1/2* variants identified in the twenty ovarian cancer patients, following the ClinVar annotation.

Sample	Type of Alteration	Gene	Protein	dbSNP	HRD Academic Score (R + W)	HRD Myriad (Score)
1	c.4284dupT	*BRCA2*	p.(Gln1429SerfsTer9)	rs80359439	**HRD-H**	D (59)
2	c.2808_2811del	*BRCA2*	p.(Ala938ProfsTer21)	rs80359351	**HRD-M**	D (54)
3	c.5266dup	*BRCA1*	p.(Gln1756ProfsTer74)	rs80357906	**HRD-H**	D (69)
4	c.649del	*BRCA1*	p.(Ser217fs)	rs878854963	**HRD-H**	D (74)
5	c.181T>G	*BRCA1*	p.(Cys61Gly)	rs28897672	**HRD-H**	D (61)
6	c.8754+4A>G	*BRCA2*	p.(Gly2919ValfsTer4)	rs81002893	**HRD-H**	D (56)
7 **^§^**	c.9052_9057del	*BRCA2*	p.(Ser3018_Lys3019del)	rs786202063	HRD-N	N (22)
8 °	c.5407-1G>A	*BRCA1*	IVS22-1G>A	rs80358029	**HRD-H**	D (56)
9	c.1984del	*BRCA1*	p.(His662ThrsfsTer39)	NR	**HRD-H**	D (64)
10	c.4284dup	*BRCA2*	p.(Gln1429SerfsTer9)	rs80359439	**HRD-M**	D (50)
11	c.6468_6469delTC	*BRCA2*	p.(Gln2157fs*)	rs80359596	**HRD-H**	D (61)
12	c.8755-1G>A	*BRCA2*	IVS21-1G>A	rs81002812	HRD-N	N (14)
13	c.1138_1177del	*BRCA1*	p.(Gln380Ter)	rs397508840	**HRD-H**	D (95)
14	NA	*NA*	NA	-	**HRD-M**	D (51)
15	c.6397dup	*BRCA2*	p.(Ser2133fs)	rs431825342	**HRD-H**	D (66)
16	c.2835dup	*BRCA1*	p.(Ile946Tyrfs)	rs80357519	**HRD-H**	D (57)
17	exon 20del	*BRCA1*	-	NR	ND	D (45)
18	c.3253dup	*BRCA1*	p.(Arg1085fs)	rs80357517	**HRD-H**	D (67)
19	c.65T>C	*BRCA1*	p.(Leu22Ser)	rs80357438	**HRD-H**	N (41)
20	c.2350_2351del	*BRCA1*	p.(Ser784ValfsTer5)	rs397508960	**HRD-M**	D (52)

**NA** = absence of *BRCA1/2* alterations; **^§^** VUS; ° Endometrioid ovarian cancer; **D** = Deficient; **N** = HRD-Negative; **ND** = undetermined, due to the low quality of DNA; **NR:** not reported within all databases.

**Table 2 ijms-24-17095-t002:** Patient stratification by GIA score, calculated using the R and W methods. HRD-Deficient (GIA > 20), HRM-Mild (15 < GIA < 19) and HRN-negative (GIA < 14).

	GIA Using “W-Assay”			
**GIA using “R-assay**”		**HRN**	**HRM**	**HRD**
	**HRN**	3	0	0
	**HRM**	1	3	0
	**HRD**	2	0	11
	Cohen’s kappa (k) = 0.736 (*p* = 0.000)			

**Table 3 ijms-24-17095-t003:** Patient stratification in 2 groups using GIA score calculated using the R and W sWGS methods. HRD-Deficient (GIA > 15) and HRN-negative (GIA < 14).

	**GIA Using Myriad**		
**GIA using “R assay**”		**HRN**	**HRD**
	**HRN**	2	1
	**HRD**	1	16
	**GIA by Myriad**		
**GIA using “W assay**”		**HRN**	**HRD**
	**HRN**	2	2
	**HRD**	1	15

**Table 4 ijms-24-17095-t004:** Correlation of aHRD-MITO classifications and Myriad scores.

	Myriad ≤ 42	43 < Myriad < 55	Myriad ≥ 56
**aHRD-MITO**	**HRN**	**HRM**	**HRD**
	**(GIA < 14)**	**(GIA 15–19)**	**(GIA > 20)**

**Table 5 ijms-24-17095-t005:** Patients’ clinical features.

Clinical Features	N	(%)
**Age**		
Median (range)	59 (41–81)	-
**Age at diagnosis**		
Median (range)	56.50 (40–79)	-
**Histology**		
High-grade serous carcinoma	19	95
Endometrioid carcinoma	1	5
**Grade (G)**		
G3	20	
**FIGO stage**		
IA	1	5
IIIA1	3	15
IIIB	1	5
IIIC	11	55
IV	1	5
IVA	1	5
IVB	2	10
**Therapy**		
Neoadjuvant	6	
Adjuvant	14	
**Overall Survival (OS)**		
Median Days (range)	660.5 (296–4369)	-
**Progression-Free Survival (PFS)**		
Median Days (range)	547 (222–4061)	-

## Data Availability

Due to the privacy policy, dataset cannot be shared.

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
