# Peer review of "Homologous Recombination Deficiency (HRD) Scoring, by Means of Two Different Shallow Whole-Genome Sequencing Pipelines (sWGS), in Ovarian Cancer Patients: A Comparison with Myriad MyChoice Assay"

_ijms, 2023, doi:10.3390/ijms242317095_

Round 1
Reviewer 1 Report
Comments and Suggestions for Authors
The authors of the manuscript entitled “Homologous recombination deficiency (HRD) scoring, by means of two different shallow whole genome sequencing pipelines (sWGS), in ovarian cancer patients: comparison with Myriad MyChoice assay” describe an alternative method for determining the HRD status of ovarian cancer. To do this, they use a nanopore sequencing technique with low coverage and an evaluation algorithm that appears to allow clear classification. It’s not clear whether the collection of HRD status is done without taking LOHs into account. If so, this process would circumvent Myriad's patent protection and the reviewer congratulates the authors on this promising result. This should be made clearer.
The language used for the abbreviation “HRN” should be adjusted. HRD mediates a deficiency in homologous recombination. Although the authors mean a negative HRD finding when using “HRN”, homologous recombination is not negative in this case. Rather, homologous recombination is very much “capable” in this case. I therefore recommend changing HRN to HRC.
Furthermore, there is no indication as to whether the patients have already received PARPi. Figure 1 shows PFS times of up to 150 (=12.5 years) months. However, PARPi was first approved 9 years ago. The Kaplan-Meier curve could therefore be based on different treatments, which should be made clear here.
Further small adjustments are necessary:
Line 1: “Title” should be deleted.
Line 189: The definition of HRM is not clear. How are the values GIA=14 and GIA=20 classified and do they apply equally to R and W?
Line 356, 403 and Table 5: Here I notice a discrepancy in how many HGSOC were included in the study.
Line 403: There are no data or images about the OS.
Line 443: “Pleas add” should be deleted.
Author Response
REV1- RESPONSE
MAJOR COMMENTS
The authors of the manuscript entitled “Homologous recombination deficiency (HRD) scoring, by means of two different shallow whole genome sequencing pipelines (sWGS), in ovarian cancer patients: comparison with Myriad MyChoice assay” describe an alternative method for determining the HRD status of ovarian cancer. To do this, they use a nanopore sequencing technique with low coverage and an evaluation algorithm that appears to allow clear classification. It’s not clear whether the collection of HRD status is done without taking LOHs into account. If so, this process would circumvent Myriad's patent protection and the reviewer congratulates the authors on this promising result. This should be made clearer.
Response: We thank the reviewer for these appropriate comments and we confirm as we did not include either the LOH and other signatures, like the presence of a pathogenic variant in BRCA1/2 within our algorithm. Our bioinformatics pipeline has been better detailed in the revised version of our manuscript (please, see M&M section).
The language used for the abbreviation “HRN” should be adjusted. HRD mediates a deficiency in homologous recombination. Although the authors mean a negative HRD finding when using “HRN”, homologous recombination is not negative in this case. Rather, homologous recombination is very much “capable” in this case. I therefore recommend changing HRN to HRC.
Response: We considered as NEGATIVE samples that do not show significant large rearrangements or structural alterations. We are not in agreement with the reviewer due to the fact that the HRD capability or proficiency are biological and functional concepts: it has been published in many studies that also patients that are non HRD can respond to the PARP-inhibitors. This means that the biology of HRD is far to be completely deciphered. We have added a specific comment to this regard in the revised paper.
Furthermore, there is no indication as to whether the patients have already received PARPi. Figure 1 shows PFS times of up to 150 (=12.5 years) months. However, PARPi was first approved 9 years ago. The Kaplan-Meier curve could therefore be based on different treatments, which should be made clear here.
Response: This is true, but we have to consider that patients treated before PARP-i approval were surgically debulked and treated with platinum-based drug. Only about nine years ago (2014) the PARPi were introduced as maintenance therapy for gBRCA1/2 mutated high grade serous epithelial ovarian cancer patients. Generally patients better responding to platinum drugs are more sensitive to PARPi. Therefore this PFS corresponds to the entire period of treatment and follow-up, as specified in line 455 (” PFS was measured from date of diagnosis to the date of earliest recurrence, progression, or death”).
English has been reviewed.
Further small adjustments are necessary:
Line 1: “Title” should be deleted.
Response: Done
Line 189: The definition of HRM is not clear. How are the values GIA=14 and GIA=20 classified and do they apply equally to R and W?
Response: We have specified within M&M section how we defined the HRM , the latter corresponding the 15<LGA<19. This categorization was used for both the R and W sWGS chemistries.
Line 356, 403 and Table 5: Here I notice a discrepancy in how many HGSOC were included in the study.
Response: Thanks a lot. We have removed the typo within Table5.
Line 403: There are no data or images about the OS.
Response: Data have been already reported in Table 5.
Line 443: “Pleas add” should be deleted.
Response: DONE

Reviewer 2 Report
Comments and Suggestions for Authors
Dear authors
Thank you for submitting your draft titled "Homologous recombination deficiency (HRD) scoring, by means of two different shallow whole genome sequencing pipelines (sWGS), in ovarian cancer patients: comparison with Myriad MyChoice assay."
My main concern are related with the issue that all your conclusions are based on a single statistical analysis with a high degree of bias, without performing any other more robust statistical test or that complements Cohen's k test..
Due to its small sample size and asymmetric proportion of data, the effects of potential bias, may result in spurious associations and misinterpretation with statistical significance in an erroneous manner.
It is suggested strengthen your statistical strategy redesign according to the suggested recommendations to reduce the probability of bias and type I and type II errors.
Annexed in the .pdf document my specific comments and recommendations to carry out the analysis of your variables again.
We invite you to take into consideration our comments and suggestions in the attached .pdf to materialize substantial improvements and resubmit them for consideration for publication.

Author Response
My main concern are related with the issue that all your conclusions are based on a single statistical analysis with a high degree of bias, without performing any other more robust statistical test or that complements Cohen's k test. Due to its small sample size and asymmetric proportion of data, the effects of potential bias, may result in spurious associations and misinterpretation with statistical significance in an erroneous manner. It is suggested strengthen your statistical strategy redesign according to the suggested recommendations to reduce the probability of bias and type I and type II errors. Annexed in the .pdf document my specific comments and recommendations to carry out the analysis of your variables again.
We invite you to take into consideration our comments and suggestions in the attached .pdf to materialize substantial improvements and resubmit them for consideration for publication.
RESPONSE: We thank the reviewer for the comments and advices. In regard to the concern about Cohen's kappa that is the most appropriate statistical test for methods comparison, we underline as our paperi s focused on the evaluation of the degree of agreement between two sWGS pipelins copupled with the same bioinformatic algorithm, wich was academically developed in a previous research. We are showing as, regardless of the sWGS pipeline, our bioinformatic tool produce similar and superimposable results in terms of HRD assessment, as also coinfirmed when compared to Myriad myChoice test. The Fisher exact test is not relevant to this situation. Finally, also we also performed a correlation test with the Spearman's correlation coefficient, that is reported in material methods and results section. Here there is the guideline that we used to select the most appropriate test https://doi.org/10.1016/j.theriogenology.2010.01.003
Round 2
Reviewer 2 Report
Comments and Suggestions for Authors
Dear authors
We apreciate your efforts to improve your paper. In general, most of our recommendations and suggestions were taken on board and corresponding modifications were made in the new manuscript.
However, this is the problem with giving much importance in your discussion and conclusions to Cohen's kappa statistic as single statistical tool.
Although having your counts analyzed for comparison in a cross table, the kappa statistic only estimates the evaluation agreement (how similar the data are). We recommend using at least one other statistical test. From our perspective, a test of association, using a Fisher's exact test, would expect to not find a significant difference between the techniques.
Since a letter with a response fletter and arguments against is not added, if you decide not to do this to have more evidence, we would like to know their specific arguments.
Author Response
Please find in the attached form the response. Thank you for the suggestions.
Regards.
EC

Round 3
Reviewer 2 Report
Comments and Suggestions for Authors
Dear authors
Thank you for submitting your draft modifications titled *Homologous recombination deficiency (HRD) scoring, by means of two different shallow whole genome sequencing pipelines (sWGS), in ovarian cancer patients: comparison with Myriad MyChoice assay.*. I appreciate a lot your effort. My main concern related to the methodology has been clarified and the drafting was improved as well as editing based in my recommendtions and found additional material to improve the article . I think the material can be published.
Greetings from México.
Raúl C. Baptista Rosas MD PhD
Universidad de Guadalajara